# Rapid CO$_2$ mineralisation into calcite at the CarbFix storage site quantified using calcium isotopes

Philip A.E. Pogge von Strandmann [1], Kevin W. Burton[2], Sandra O. Snæbjörnsdóttir[3], Bergur Sigfússon[4], Edda S. Aradóttir[4], Ingvi Gunnarsson[4], Helgi A. Alfredsson[3], Kiflom G. Mesfin[3,5], Eric H. Oelkers[1,3,6] & Sigurður R. Gislason[3]

The engineered removal of atmospheric CO$_2$ is now considered a key component of mitigating climate warming below 1.5 °C. Mineral carbonation is a potential negative emissions technique that, in the case of Iceland's CarbFix experiment, precipitates dissolved CO$_2$ as carbonate minerals in basaltic groundwater settings. Here we use calcium (Ca) isotopes in both pre- and post-CO$_2$ injection waters to quantify the amount of carbonate precipitated, and hence CO$_2$ stored. Ca isotope ratios rapidly increase with the pH and calcite saturation state, indicating calcite precipitation. Calculations suggest that up to 93% of dissolved Ca is removed into calcite during certain phases of injection. In total, our results suggest that 165 ± 8.3 t CO$_2$ were precipitated into calcite, an overall carbon storage efficiency of 72 ± 5%. The success of this approach opens the potential for quantification of similar mineral carbonation efforts where drawdown rates cannot be estimated by other means.

---

[1] LOGIC, Institute of Earth and Planetary Sciences, University College London and Birkbeck, University of London, Gower Street, London WC1E 6BT, UK. [2] Department of Earth Sciences, University of Durham, Durham DH1 3LE, UK. [3] Institute of Earth Sciences, University of Iceland, Reykjavik 600169-2039, Iceland. [4] Reykjavik Energy, Bæjarhálsi 1, Reykjavik 551298-3029, Iceland. [5] HS Orka, Svartsengi, 240 Grindavik, Iceland. [6] GET, CNRS UMR-5563, 14, Avenue Édouard Belin, Toulouse 31400, France. Correspondence and requests for materials should be addressed to P.A.E.P.v.S. (email: p.strandmann@ucl.ac.uk)

The engineered removal of $CO_2$ from the atmosphere (negative emissions) is now thought to be crucial for keeping global warming under the limit of 1.5 °C[1]. A plethora of different potential geo-engineering techniques has been proposed, with different costs, potential and environmental consequences of implementation[2]. Indeed, the overall scale of necessary carbon capture and storage (CCS) suggests that a combination of different techniques would be most practical. An effective and, critically, efficient approach is the enhancement of natural carbon storage processes, in particular those associated with the long- and short-term carbon cycle, including carbonate formation, alkalinity drawdown and organic carbon enhancement[3,4].

Of these, one particular method is mineral carbonation, which is the engineered enhancement of carbonate precipitation and hence $CO_2$ storage. The CarbFix programme in Iceland has now progressed from a pilot (original CarbFix)[5] to the industrial scale (CarbFix2)[6]. In simple terms, $CO_2$ dissolved in water is injected into natural basaltic aquifers near the Hellisheidi geothermal power plant. Basalt, which at this site contains ~10 wt% CaO[7] plus other carbonate-forming cations such as Mg or Fe, is highly reactive. The concept is that the precipitation of carbonates at depth occurs from carbon sourced from the $CO_2$ injection, and from calcium (or Mg or Fe) sourced from dissolution of the surrounding basalts. In the pilot experiment, carbonate precipitated within 2 years of injection at 20–50 °C and a few months at 60 to 260 °C[5,6,8]. This technique has the advantage that carbon is stored as solid carbonate minerals, a significantly more robust substrate than supercritical $CO_2$ stored via other CCS techniques, and that this mineralised carbon is relatively stable over long timescales[5,9].

Calculation of the carbon stored during $CO_2$ injection at the pilot CarbFix site was made through the use of conservative (non-reactive) and reactive tracers, which were added to the injected $CO_2$-charged water (including radioactive $^{14}C$)[5,8]. Carbon storage within carbonate minerals was calculated by determining the difference between the amount of injected dissolved inorganic carbon (DIC), $^{14}C$ and conservative tracers, with that sampled from a downstream monitoring well[5,8]. The stability of different minerals (different carbonates, as well as silicate secondary minerals) through the subsurface sequestration site has also been modelled, including prediction of groundwater conditions (temperature, pH, concentration) under which carbonate will form[7,8,10]. However, monitoring the extent and flux of carbonate and other secondary mineral precipitation using dissolved elemental concentrations alone is difficult because supply rates are not readily quantified, and, moreover, there are no diagnostic elemental signals that can uniquely indicate which metal-bearing phases have precipitated. Using concentration of solids from drill cores would also be extremely difficult, due to the relatively small amount of solid precipitated over millions of $m^3$ and hundreds of Mt of basaltic rock. This highlights the difficulty of determining carbonation efficiency in field studies compared to laboratory experiments (e.g. ref. [11]).

Over recent years the stable isotopes of a number of elements have been used with considerable success as tracers of weathering processes in groundwaters and rivers. Calcium isotopes (here reported as $\delta^{44/40}Ca$ relative to the standard NIST (National Institute of Standards and Technology) 915a) have an obvious potential as a tracer of carbonate precipitation processes. There is no known isotope fractionation associated with the dissolution of primary minerals, but there is significant fractionation associated with the precipitation of secondary Ca-rich phases[12]. These are dominated by carbonates, but also include minerals such as Ca-smectites and -zeolites. In all cases precipitation leaves waters enriched in the heavy isotopes[12–16]. Therefore, calcium isotopes

have the potential to serve as a diagnostic tracer of carbonate precipitation rates (and hence $CO_2$ sequestration rates) at the CarbFix site.

Here we present Ca isotope data for the pre- and post-carbon injection groundwaters collected from the original CarbFix pilot site, to determine whether carbonate formation (and hence $CO_2$ sequestration) rates can be directly quantified. Ca isotope ratios increase with pH and calcite saturation states, indicating rapid calcite precipitation. The most rapid calcite mineralisation and $CO_2$ drawdown occurred within 1–2 months of the cessation of carbon injection. Overall, the Ca isotopes show that $165 \pm 8.3$ ($2\sigma$) t $CO_2$ of the 220 t injected were precipitated into calcite, confirming calcite as the primary carbon storage phase, and giving a carbon storage efficiency of $72 \pm 5\%$.

## Results

**The CarbFix site.** The location of the original CarbFix pilot-study injection site is in SW Iceland, ~30 km east of Reykjavik, and 3 km SW of the Hellisheidi geothermal power plant, operated by Reykjavik Energy. The injection site has eight monitoring wells ranging from 50 to 2000 m depth (Supplementary Figs. 1–3)—six of these are located downstream from the HN-02 injection well. The shallow wells source water from 200 to 300 m depth, above a low-permeable hyaloclastite formation, while the deeper wells are encased to 400 m depth and sample the target formation (Supplementary Figs. 1–3). The injection site and its groundwaters have been characterised[7], and a number of wells were sampled multiple times (up to three) over 3 years (2008–2010) prior to the $CO_2$ injection.

During the original CarbFix injection, water was pumped from well HN-01, and co-injected with $CO_2$ into well HN-02. This occurred in two phases: first, starting in late January 2012, 175 t of pure $CO_2$ was injected together with the water[17]. This phase ran continuously until 9 March 2012; the second phase was from mid-June to early August 2012, in which 73 t of a gas mixture containing 75 mol% $CO_2$, 24 mol% $H_2S$ and 1 mol% $H_2$ was injected. The plume of injected material was monitored at the closest monitoring well HN-04, 500 m from HN-02 at depth. Changes in DIC, pH and tracer concentration were observed starting within 2 weeks of injection. Tracer injection suggests that ~95 ± 3% of the injected carbon was mineralised within 2 years[5,8]. The rapid conversion of the dissolved $CO_2$ to carbonate minerals is likely due to the novel method of dissolved $CO_2$ injection, the rapid dissolution rate of basalt, providing the necessary cations, such as Ca, Mg or Fe, the mixing of the injected water with alkaline groundwaters and the dissolution of pre-existing carbonates at the onset of $CO_2$ injection[5]. Analysis of solids recovered from the monitoring well and pumps shows the precipitation of calcite, but no aragonite precipitation[5].

**Groundwater Ca isotope results.** We report Ca isotope ratios from 13 pre-injection aquifer samples characterising the injection site, including both shallow and deep groundwaters. Following this, we also report Ca isotope ratios from a post-injection time series of 19 samples from a single monitoring well. Calcium concentrations range from 1.2 to 21.7 μg/ml in the pre-injection samples[7], and from 1.4 to 16.3 μg/ml in the post-injection samples[8]. All data are presented in Tables 1 and 2. Basalts have calcium isotope ratios of $\delta^{44/40}Ca$ ~0.8‰[14,15]. Icelandic rivers have $\delta^{44}Ca$ values of 0.9–1.3‰, and previously measured groundwaters have values of 0.51–1.5‰[14,15]. In both cases, the CarbFix site groundwaters extend to heavier Ca isotope ratios. The shallow pre-injection groundwaters have relatively low $\delta^{44}Ca$ values, and also higher Ca/Na ratios than the deeper groundwaters (Fig. 1).

**Table 1 Pre-injection conditions, elemental concentrations (from Alfredsson et al.[7]) and Ca isotope ratios**

| Borehole | Sample | Sampling date | Depth | T (°C) | pH | Na (µg/ml) | Si (µg/ml) | Ca (µg/ml) | Sr (ng/ml) | Li (ng/ml) | Calcite SI | $\delta^{44/40}$Ca | 2 s.e. |
|---|---|---|---|---|---|---|---|---|---|---|---|---|---|
| HK-7b | 08HAA08 | 08/07/2008 | Shallow | 12.4 | 7.65 | 9.17 | 12.0 | 11.3 | 20.1 | | −1.13 | | |
| HN-4 | 08HAA01 | 01/07/2008 | Deep | 32.3 | 9.43 | 48.6 | 25.1 | 1.64 | 1.62 | 0.40 | 0.12 | 1.95 | 0.05 |
| HK-34 | 08HAA03 | 04/07/2008 | Deep | 25 | 9.63 | 49.6 | 22.7 | 2.17 | 3.98 | | | | |
| HK-31 | 08HAA05 | 04/07/2008 | Deep | 17.4 | 9.29 | 41.8 | 20.7 | 3.75 | 11.9 | 0.31 | 0.29 | 2.04 | 0.04 |
| HK-26 | 08HAA06 | 04/07/2008 | Deep | 18.8 | 8.44 | 72.5 | 17.8 | 5.72 | 28.3 | | | | |
| HK-26 | 09HAA17 | 29/05/2009 | | 16.5 | 8.51 | 77.0 | 16.7 | 5.36 | 26.5 | 0.59 | 0.02 | 1.22 | 0.05 |
| HK-12 | 09HAA18 | 29/05/2009 | Shallow | 5.3 | 8.33 | 8.82 | 8.63 | 8.39 | 14.8 | 0.16 | −0.78 | 1.11 | 0.05 |
| HK-31 | 09HAA19 | 29/05/2009 | Deep | 18.9 | 9.41 | 48.6 | 21.1 | 3.69 | 11.9 | 0.44 | 0.37 | 1.91 | 0.05 |
| HK-25 | 09HAA20 | 29/05/2009 | Shallow | 7.4 | 8.09 | 8.00 | 9.0 | 8.02 | 13.9 | 0.28 | −1.00 | 1.09 | 0.04 |
| HK-34 | 09HAA21 | 29/05/2009 | Deep | 27.5 | 9.79 | 55.0 | 24.6 | 1.30 | 1.74 | 0.23 | 0.15 | 2.04 | 0.03 |
| rpt | | | | | | | | | | | | 1.98 | 0.06 |
| HK-7b | 09HAA22 | 29/05/2009 | Shallow | 11.7 | 7.67 | 12.2 | 13.0 | 21.7 | 38.9 | 0.31 | −0.50 | 1.14 | 0.03 |
| HN-1 | 09HAA23 | 29/05/2009 | Deep | 24.7 | 9.26 | 43.5 | 14.1 | 5.37 | 18.1 | 0.16 | 0.51 | 2.07 | 0.04 |
| HN-4 | 09HAA24 | 05/06/2009 | Deep | 34.5 | 9.56 | 55.6 | 25.2 | 1.32 | 1.27 | 0.27 | 0.10 | 1.89 | 0.04 |
| HK-26 | 10HAA26 | 25/06/2010 | Deep | 17.2 | 8.65 | 74.5 | 16.6 | 5.25 | 26.1 | 0.32 | 0.05 | 1.62 | 0.04 |
| HK-31 | 10HAA28 | 25/06/2010 | Deep | 16.9 | 9.55 | 48.6 | 23.1 | 3.61 | 12.0 | 0.41 | 0.42 | 1.91 | 0.06 |
| HN-4 | 10HAA29 | 25/06/2010 | Deep | 30.4 | 9.69 | 54.7 | 30.6 | 1.29 | 1.14 | | | | |
| HK-34 | 10HAA30 | 25/06/2010 | Deep | 28.0 | 9.88 | 51.0 | 27.1 | 1.21 | 1.67 | 0.24 | 0.03 | 1.75 | 0.03 |

Borehole names are the same as in Figs. S1–3. The shallow wells, HK-7b, HK-12, and HK-25, draw water from the dominate aquifers in the upper system, whereas the deeper wells, HN-2, HN-4, HK-34, HN-1, HK-31, and HK-26, draw water from the highest discharge aquifers below 400 m since these wells are cased down to that depth[7]

**Table 2 Post-injection conditions, elemental concentrations (from Snæbjörnsdóttir et al.[8]) and Ca isotope ratios**

| Sample | Sampling date | pH | Na (mg/ml) | Ca (µg/ml) | Sr (ng/ml) | Li (ng/ml) | Calcite SI | $\delta^{44/40}$Ca | 2 s.e. |
|---|---|---|---|---|---|---|---|---|---|
| 12KGM08 | 09/02/2012 | 8.98 | 50.8 | 5.44 | 3.38 | 0.28 | 0.44 | 1.54 | 0.03 |
| 12KGM11 | 16/02/2012 | 7.94 | 52.0 | 7.81 | 4.67 | 0.29 | −0.28 | 1.12 | 0.05 |
| 12KGM19 | 27/02/2012 | 7.18 | 54.3 | 9.56 | 5.62 | 0.31 | −0.88 | 1.05 | 0.04 |
| 12KGM25 | 08/03/2012 | 6.79 | 53.2 | 14.4 | 8.58 | 0.35 | −1.07 | 0.89 | 0.05 |
| 12KGM33 | 26/03/2012 | 6.71 | 54.6 | 16.3 | 10.0 | 0.36 | −1.08 | 1.09 | 0.04 |
| 12KGM44 | 18/04/2012 | 7.70 | 54.6 | 12.0 | 7.62 | 0.35 | −0.28 | 1.29 | 0.04 |
| 12KGM49 | 04/05/2012 | 9.00 | 55.8 | 6.50 | 4.23 | 0.32 | 0.57 | 2.13 | 0.06 |
| 12KGM60 | 30/05/2012 | 8.81 | 53.7 | 8.75 | 5.87 | 0.32 | 0.57 | 2.11 | 0.04 |
| 12SOS01 | 28/06/2012 | 7.36 | 55.0 | 11.9 | 8.14 | 0.36 | −0.60 | 1.07 | 0.04 |
| 12SOS09 | 17/07/2012 | 8.28 | 55.8 | 10.9 | 7.86 | 0.36 | 0.25 | 1.91 | 0.05 |
| rpt | | | | | | | | 1.94 | 0.04 |
| 12SOS15 | 31/07/2012 | 8.32 | 55.9 | 11.3 | 8.13 | 0.35 | 0.37 | 2.13 | 0.05 |
| rpt | | | | | | | | 1.99 | 0.06 |
| 12SOS21 | 14/08/2012 | 7.25 | 57.0 | 14.3 | 9.96 | 0.43 | −0.79 | 0.80 | 0.05 |
| 12SOS28 | 28/08/2012 | 7.50 | 57.9 | 13.1 | 9.57 | 0.44 | 0.02 | 1.64 | 0.06 |
| 12SOS34 | 24/09/2012 | 8.23 | 60.5 | 12.1 | 9.53 | 0.36 | 0.29 | 2.18 | 0.02 |
| 12SOS39 | 29/10/2012 | 8.26 | 61.6 | 11.0 | 9.45 | 0.46 | 0.22 | 2.18 | 0.04 |
| 13SOS01 | 07/01/2013 | 8.73 | 62.7 | 6.85 | 7.42 | 0.42 | 0.41 | 2.02 | 0.04 |
| 13SOS10 | 16/04/2013 | 8.76 | 62.4 | 7.19 | 7.38 | 0.44 | 0.47 | 2.10 | 0.03 |
| 13SOS17 | 10/06/2013 | 8.86 | 60.4 | 7.13 | 7.43 | 0.56 | 0.54 | 1.54 | 0.03 |
| 14SOS11 | 17/03/2014 | 9.08 | 63.7 | 3.49 | 4.12 | 0.56 | 0.31 | 1.39 | 0.05 |

These samples are a time series from the monitoring well HN-4 with a constant temperature of 35 °C

Pre-injection samples have $\delta^{44/40}$Ca values of 1.09 to 2.07‰ (relative to SRM-915a), and exhibit a positive relationship with pH (Fig. 2). The post-injection $\delta^{44/40}$Ca values exhibit similar overall isotope ratios of 0.8–2.1‰, but have considerably more scatter than the samples from the monitoring well before the $CO_2$ injection (1.89–1.95‰ in HN-04). The post-injection samples also show a positive co-variation with pH (Fig. 2).

## Discussion

Prior to the $CO_2$ injection, the Ca isotopic compositions of the aquifers show a significant variation (1.09 to 2.07‰ relative to SRM-915a), trending from similar to somewhat higher values than observed in Icelandic rivers[14,15] (Fig. 1). These studies suggest that rivers' $\delta^{44}$Ca are largely controlled by mixing between the dissolution of basalts ($\delta^{44}$Ca $\sim$ 0.8‰) and calcite (which is undersaturated and hence dissolving) plus hydrothermal and meltwater inputs. Additional isotope fractionation is caused by the formation of Ca-bearing secondary minerals, such as heulandite or stilbite.

Fractionation processes can be examined by comparing Ca isotope ratios to Ca/Na ratios. In basaltic environments, Na is the most mobile of the major cations[18], consequently it most readily goes into solution compared to less mobile cations, which are preferentially taken up by secondary minerals. Only the shallow groundwaters have generally similar Ca/Na values to Icelandic rivers (Fig. 1). In contrast, the deeper aquifers tend to have lower pre-injection Ca/Na, suggesting that relatively more Ca is being

removed from these waters into secondary phases than in most rivers. In these groundwaters there is no correlation between $\delta^{44}$Ca and the calculated saturation index of zeolites such as heulandite[8,10,19]. Equally, there is no co-variation between $\delta^{44}$Ca and Mg isotope ratios, where the latter are thought to be controlled by the precipitation of smectites[20]. This lack of correlation also suggests that any precipitating secondary silicates are Mg-rich, but Ca-poor. However, an additional process that is highly likely to be controlling Ca isotopes in these waters is the precipitation of carbonate, especially in the higher pH aquifers. Calcite has been observed precipitating as a secondary mineral from these pre-injection waters[7,10], as well as in other basaltic settings[21], and indeed the precipitation of carbonate is the main goal of the CarbFix project.

There is a strong relationship between calcite saturation (calculated using PHREEQC[8]) and $\delta^{44}$Ca (Fig. 2b), largely controlled by a similar co-variation between $\delta^{44}$Ca and pH (Fig. 2a). Importantly, when calcite is undersaturated, $\delta^{44}$Ca is low (close to basaltic values, and similar to Icelandic rivers[14,15]), which likely represents secondary Ca-silicate precipitation. In contrast, when calcite is

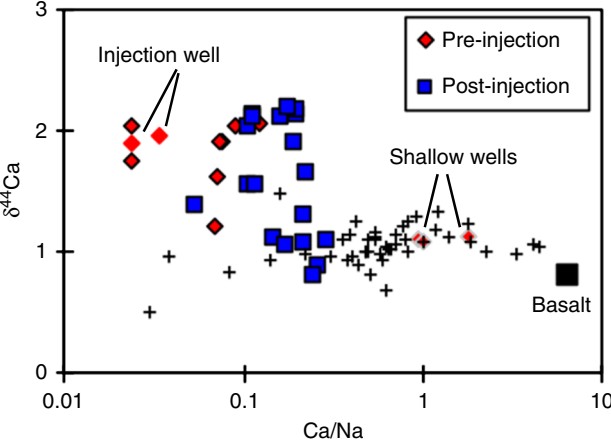

**Fig. 1** Ca isotopes as a function of Ca/Na ratios. Elemental/Na ratios theoretically represent the uptake of the element in question by secondary minerals, relative to the highly mobile Na cation. The red diamonds with grey outlines represent the shallow groundwaters, while the pure red diamonds represent the injection well. Black crosses are previously measured Icelandic rivers and groundwaters[14,15]. Pre-injection samples are from a range of monitoring wells, while post-injection samples are a time series from a single well (HN-4)

supersaturated, and therefore likely precipitating, $\delta^{44}$Ca in the waters rapidly increases. Calcite is known to preferentially take up light Ca isotopes. The precise value for the fractionation factor is debated, but is around 0.6–0.8‰ ($\alpha = 0.999$–0.9995) for inorganic calcite, with a degree of temperature-dependence[13,22]. Using the relationship from Gussone et al.[7] and the measured temperatures, the pre-injection samples are likely to have fractionation factors into calcite of ~0.9991–0.9995, due to temperatures ranging from 5 to 35 °C. Using a standard Rayleigh fractionation equation and assuming that all Ca initially originates from isotopically conservative basalt dissolution (this method is detailed in the Methods, and has previously been used to calculate similar parameters in speleothem systems[16]; equilibrium fractionation using this assumption cannot reproduce the higher $\delta^{44}$Ca values), it is calculated that around 28% of Ca is taken up into calcite from solution at pH < 8.5, but this reaches ~90% at pH 9.5–10. In other words, at high pH, most of the dissolved Ca is taken up by calcite precipitation, following these calculations.

The CarbFix injection has a clear effect on Ca isotopes: the pre-injection waters from well HN-04 (the primary monitoring well) had $\delta^{44}$Ca values of 1.89–1.95‰ between 2008 and 2009. In comparison, the same well exhibits $\delta^{44}$Ca values of 0.8–2.1‰ post-injection. The effect of the CO$_2$ addition in both injection phases was initially to decrease pH by over 2 units[8]. Following this, pH recovered (Fig. 3), because the dissolution of the host basalts and fluid mixing neutralised the pH of the injected fluids[8], allowing the precipitation of calcite[5,8] (but no observed aragonite[5]). Hence, there are two periods of calcite undersaturation, matching the two phases of injection. For the rest of the post-injection period, calcite is supersaturated, and is demonstrably precipitating on and within the pumps and pipes in the monitoring well, and is the only recovered phase containing the injected radioactive carbon ($^{14}$C)[5,8].

Similar to the pre-injection samples, the post-injection $\delta^{44}$Ca values correlate with calcite saturation (Fig. 2). Due to an increased groundwater flow rate during injection (see Methods), water and elemental fluxes are significantly higher in post-compared to pre-injection waters. Assuming that the Ca isotope ratios in the post-injection samples are dominantly controlled by calcite precipitation, identical calculations can be performed as for the pre-injection aquifers. Due to the high reactivity and high Ca concentrations of basalt, the vast majority of the calcium (and other cations necessary for carbonate precipitation, such as Mg or Fe) are supplied from the dissolution of basalt. As such, the Ca dissolved in pre-injection groundwaters is insignificant for the purposes of enhanced carbonate precipitation and the following

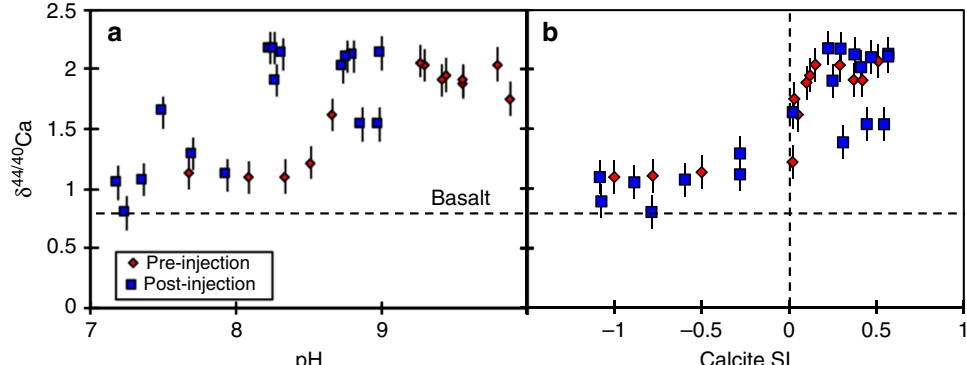

**Fig. 2** Relationship between Ca isotopes and groundwater pH and calcite saturation indices. **a** Ca isotope ratios plotted against water pH; **b** and against the calcite saturation index. Both plots show that Ca isotope ratios increase as the conditions for calcite precipitation improve. Pre-injection waters are the red diamonds, and post-injection waters the blue squares. The horizontal dashed black line represents the Ca isotope ratio of basalt, while the vertical dashed black line represents saturation (SI = 0). The error bars represent 2 s.e. internal analytical uncertainty

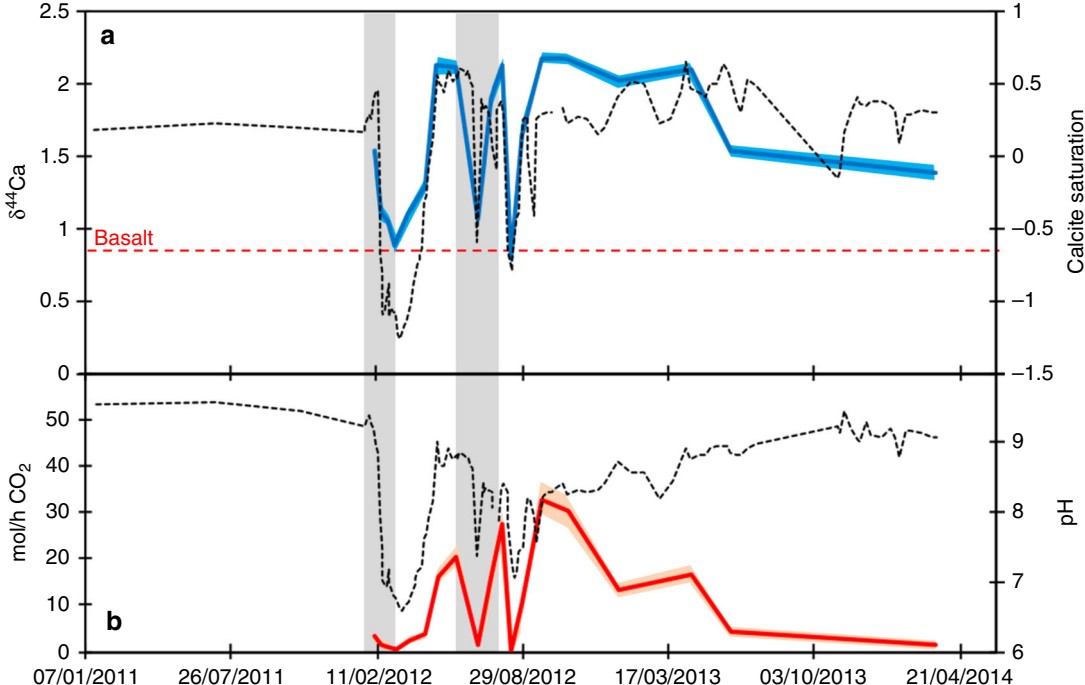

**Fig. 3** The trends of the calcite saturation state and Ca isotopes with time during the carbon injection phase from monitoring borehole HN-4. **a** Ca isotope ratios (blue line) and calcite saturation index (dotted black line). The blue shaded area represents the 2 s.e. analytical uncertainty on the isotope measurements. **b** shows the evolution of pH (dotted black line) and the calculated $CO_2$ precipitation rate based on Ca isotopes (red line). The red shaded area represents the $2\sigma$ propagated uncertainty on the individual precipitation rates. The grey shaded areas represent the carbon injection periods

mass balance calculations[5,9]. In this case, because the samples all stem from the same borehole, the temperature was constant at 35 °C[8]. Assuming that only calcite precipitation fractionated Ca isotopes towards higher values in this setting, and taking a temperature-dependent isotope fractionation factor[13,22] of 1000 ln $\alpha = -1.02 \pm 0.25 + (0.015 \pm 0.013) \times T$ (°C), this gives a fractionation factor of $\alpha = 0.9995$. The large pH variations due to the injection of $CO_2$ lead to 0 to 93% of Ca taken up into precipitated calcite. Hence, at periods of high pH, Ca storage behaviour is similar to that reported to occur to $CO_2$, in that almost 95% is sequestered into carbonates[5,8].

The precipitation rates and total mass of calcite formed during and after the CarbFix $CO_2$ injection can be determined from the initial isotope ratio of the basalt, the isotope ratios and Ca concentrations of the measured fluid samples, the fluid-calcite fractionation factor and the water flow rate. Details of this calculation are provided in the Methods. A significant advantage of determining these rates from stable metal isotope measurements in such systems is that it potentially provides information on the precipitation rates of distinct minerals simply by measuring the isotope ratios of the groundwaters and the flow rate, and without the necessity of adding tracers to the injected carbon.

In the original CarbFix experiment, approximately 230 t $CO_2$ was injected in two stages. Matter et al.[5] estimated a $CO_2$ sequestration efficiency of around $95 \pm 3\%$ based on $^{14}C$ and $98 \pm 4\%$ based on DIC. Hence, ~220 t of $CO_2$ were precipitated during the initial 2 years, with modelling suggesting this dominantly was in calcite (with modelling also suggesting that trace Fe-carbonate might form at the lowest pH[19]). To corroborate this, only calcite was found to contain the $^{14}C$ tracer[5,8]. The Ca isotopes suggest that approximately $165 \pm 8.3$ ($2\sigma$) t $CO_2$ were precipitated into calcite, a sequestration efficiency of $72 \pm 5\%$, confirming calcite as the dominant carbon mineralisation phase (Fig. 3—see Methods for error propagation formulae). The greatest precipitation rates occur within 2 months following the cessation of injection. The

estimate based on Ca isotopes is thus lower than that based on injected carbon tracers and a conservative mass balance approach[5]. In effect, this suggests that more carbon is being removed from solution than is being mineralised by calcite. This may be due to the early formation of Fe–Mg carbonate minerals over calcite at relatively low pH[19]. Overall, these results show that Ca isotopes can be used to determine the amount of $CO_2$ stored in calcite in the subsurface without the use of additional tracers.

An interesting corollary to the Ca isotope method is its ability to time-resolve calcite precipitation rates (as opposed to carbon removal rates from the injected tracers), which shows both that precipitation rates are highest 1–2 months after cessation of $CO_2$ injection, and that $\delta^{44}Ca$ values decrease towards the end of the experiment, starting about a year after the final injection phase (Fig. 3). In both cases, the observations suggest that the calcite precipitation rates are not always directly coupled to pH or calculated saturation indices (Fig. 3), likely because precipitation is driven by both fluid saturation and reactive surface area, which can vary subtly along the flowpath and with time. For example, the slower rate of precipitation near the end of the original Carbfix experiment (down to ~2 mol/h $CO_2$ compared to a height of >30 mol/h) is despite pH and calcite saturation remaining relatively high. It seems likely that this reflects the slowing of the reaction as the injected carbon (i.e. the reactants) becomes exhausted, especially given the efficiency of carbon removal from solution[8], and the system returning to its previous state (i.e. supersaturated for calcite, but with low precipitation due to low flow rates and fluxes[7]). Overall, therefore, a tracer of precipitation rates that can act in a time-resolved manner (such as Ca isotopes) can be very useful in understanding and quantifying subsurface carbon storage.

In summary, the original CarbFix industrial pilot experiment involved pumping dissolved $CO_2$ into basaltic groundwaters to precipitate carbonate, a carbon storage method known as mineral carbonation. Initial estimates, based on conservative tracers of carbon, suggested that $95 \pm 3\%$ of the added $CO_2$ (~220 t) was

removed through mineralisation. Using calcium isotopes, we estimate the amount of Ca lost from the groundwaters due to carbonate formation. The data here suggest that $165 \pm 8.3$ ($2\sigma$) t $CO_2$ were precipitated into calcite (confirming calcite as the primary carbon storage phase), a storage efficiency of $72 \pm 5\%$ ($2\sigma$). These two estimates are therefore just outside of uncertainty of each other, confirming modelling that suggests that carbon initially was mineralised into Mg–Fe carbonates at low pH. Overall, the study indicates that the efficient storage of carbon in the CarbFix experiment has been confirmed through independent methods. This method also allows a time-resolved determination of calcite precipitation rates that is distinct from pH or saturation calculations. Further, Ca isotopes appear to be a useful tool for quantifying mineral carbonation in settings where tracer injection cannot take place, when permissions (or finances) for using chemical or radioactive do not exist, or during natural weathering involving rapid carbonation.

## Methods

**Analytical methods**. Calcite SI (saturation indices) are taken from Alfredsson et al.[7] for pre-injection samples, and from Snæbjörnsdóttir et al.[8] for post-injection samples. In both cases they were calculated using the PHREEQC programme, using measured pH, temperature, DIC and ionic strengths of individual samples.

Around 10 μg of calcium was purified through a two-stage column procedure, the first column containing AG50 X12 resin, which removes most matrix elements, and the second containing Sr-spec resin, which removes any Sr. Splits collected before and after the Ca elution peak contained <0.5% of the Ca, indicating column yields of >99.5%. Analyses were performed on a Nu Instruments MC-ICP-MS at Oxford, relative to the standard SRM-915a. Sr isobaric interference was monitored at mass 43.5, and used to correct the $^{42}$Ca, $^{43}$Ca, and $^{44}$Ca intensities. This column and mass spectrometric methodology has been detailed in a series of studies[23–26]. Seawater measured by this method yielded $\delta^{44/40}$Ca values of $1.92 \pm 0.14$‰ (2 s.d., $n = 16$), in keeping with other studies[14,27].

**Precipitation calculations**. Given an initial isotope ratio (basalt dissolution provides almost all of the dissolved Ca), the isotope ratio of the measured solutions, and a fractionation factor, the fraction of an element remaining in solution relative to that taken into the solid can be calculated using a standard Rayleigh fractionation equation (Eq. 1):

$$\delta = \delta^{i} + 1000(\alpha - 1)lnf, \qquad (1)$$

where $\delta$ is the $\delta^{44/40}$Ca of the groundwater and $\delta^{i}$ is the $\delta^{44/40}$Ca of the starting compositions of the input Ca, in this case the Ca isotope ratio of basalt. $\alpha$ is the isotopic fractionation factor, and $f$ is the fraction of Ca remaining in solution[16]. This equation is solved for $f$, which allows calculation of the amount of Ca precipitated, when using the Ca concentration of the solution. This approach also has the advantage that any dissolution of pre-existing calcite, which may occur immediately after injection when pH is low (Fig. 3), will result in very high values of $f$. Hence, the values of $f = 0.84$–1.00 determined for these periods means that very little apparent calcite precipitation occurred (Supplementary Fig. 4). In fact, if these values are set to >1 (implying Ca gain from dissolution), the impact on the final amount of sequestered $CO_2$ is <0.1 t.

In turn, $f$ values can be converted to a calcite (and hence $CO_2$) precipitation rate (e.g. mol/h), by factoring in the water flow rate, according to the standard equation[28] (Eq. 2):

$$[x] = \left[ \frac{(x)_{solution}}{f} - (x)_{solution} \right] D, \qquad (2)$$

where, in this case, $[x]$ is the Ca concentration in carbonate, $(x)_{solution}$ is the concentration in the groundwater, and $D$ is the $x$/Ca partition coefficient (here =1). The flux was obtained by multiplying the concentration (in mol/kg) by the flow rate (in l/h). Carbonate and $CO_2$ precipitation were calculated according to (using here a simplified equation based on wollastonite):

$$CO_2 + H_2O + CaSiO_3 = CaCO_3 + H_2O + SiO_2.$$

This is slightly different from the normal silicate weathering equation because the pH is considerably lower in the injected medium, and hence the ocean-based carbonate cycle does not apply.

The natural groundwater flow rate was extremely slow (~25 m/yr), but, during the CarbFix experiment, water was pumped into and out of different boreholes to speed the water flow rate (input pump rate 7200 l/h; output pump rate 3500 l/h). This then allows us to calculate mineral precipitation rates. In principle, this could be a significant advantage of measuring stable metal isotopes in such systems, because it potentially provides information on the precipitation rates of different minerals simply by measuring the isotope ratios of the groundwaters and the flow rate. In all cases, the isotope ratio of the injected water was not taken into account

(well HN-1). This is because the pre-injection Ca fluxes are insignificant compared to post-injection ones, due to the increased flow rates. Ca/Na ratios also approximately an order of magnitude lower in the pre-injection waters (Fig. 1). Also, the Ca isotope ratio is significantly different from the post-injection values.

Uncertainty was propagated from a combination of analytical uncertainty (isotopic and concentration) and the uncertainty on the isotopic fractionation factor (also effectively analytical uncertainty). The propagated uncertainty was then calculated according to (Eq. 3):

$$\sigma_P^2 = (t_2 - t_1)^2 \sigma_1^2/4 + (t_n - t_{n-1})^2 \sigma_n^2/4 + \sum_{i=2}^{n-1} (t_{i+1} - t_{i-1})^2 \sigma_i^2/4, \qquad (3)$$

where $t$ is the time at the $i$th data point. Of the reported uncertainty (Fig. 3), approximately 40% is from analyses (similar to other Ca isotope studies that also calculated values of $f$[16]), and the remainder largely from uncertainty on the fractionation factor.

## Data availability

The authors declare that the data supporting the findings of this study are available within the paper (and its supplementary information files).

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

## Acknowledgements

Analyses and PPvS were funded by NERC Advanced Fellowship NE/I020571/2 and ERC Consolidator grant CONTROLPASTCO2 682760. Chris Coath (Bristol University) is thanked for discussions and formulae on error propagation. The CarbFix pilot infrastructure, injection, sampling, sample distribution and interpretations were funded by the European Commission through the projects CarbFix (EC coordinated action 283148); Min-GRO (MC-RTN-35488); Nordic fund 11029-NORDICCS; the Icelandic GEORG Geothermal Research fund (09-02-001); the U.S. Department of Energy under award number DE-FE0004847) and Reykjavík Energy. CarbFix has further received grants from the European Union's Horizon 2020 research and innovation program under grant agreements No. 764760 (CarbFix2) and 764810(S4CE). We are indebted to Matin Stute and Jenifer Hall at Columbia University and Juerg Matter at Columbia University and Southampton University; Einar Örn Thrastarson, Trausti Kristinsson, Vordis Eiriksdottir, Halldor Bergmann and Thorsteinn A. Thorgeirsson at Reykjavík Energy; Vigdis Harðardottir, Finnbogi Oskarsson, Kristjan Hrafn Sigurðsson and Steindor Nielsson at ISOR; and Thorsteinn Jonsson, Sveinbjörn Steinthorsson, Iwona Galeczka, Eydıs S. Eiriksdottir, Deirdre Clark, Chris Grimm and Flora Brocza at the University of Iceland for helping the injection and sampling campaign.

## Author contributions

P.A.E.P.v.S. carried out the analyses, interpreted the data and wrote the manuscript, with contributions from K.W.B., S.O.S., B.S., E.S.A., I.G., H.A.A., K.G.M., E.H.O. and S.R.G., who are also all involved in the CarbFix project. S.O.S. and S.R.G. provided the samples.

## Additional information

**Competing interests:** The authors declare no competing interests.

**Journal Peer Review Information**: *Nature Communications* thanks Fei Wang and other anonymous reviewers for their contribution to the peer review of this work. Peer reviewer reports are available

