## [Peer Review File · Nature Communications]

Reviewers' comments:

Reviewer #1 (Remarks to the Author):

The paper shows a new method, calcium isotope-based quantification of in-situ mineral carbonation rates. The application and development of this method is novel and will be of interest to others in the wider field (passive carbonation). However, the current-version paper is hard to understand for the readers who are interested in the in-situ and ex-situ mineral carbonation as well as passive mineral carbonation but not in the field of geology or isotope technology. I have the following suggestions and questions to clarify, detailed below:

1. The structure of this paper needs to be adjusted. The innovation in this paper is that the calcium isotope method can be used into the in-situ mineral carbonation and the quantification of CO₂ sequestration rates. The authors need to clearly clarify how this method was applied into your project and how to calculate the efficiency?
2. What are the relationships among CarbFix, CarbFix2 and CarbFix1? Is the CarbFix1 the same as CarbFix? The authors need to clarify it.
3. The description about the calculation based on the concentration needs to be reduced and concentrated. The focus in this paper is not to describe the method based on the change of concentration anymore. In contrast, the problems of the method based on the concentration needs to be clearly clarified and the method based on the calcium isotope method needs to be described in more details.
4. Line 57 - 59: what are the "different minerals", such as? Why do the authors need to mention the modelling about the different minerals?
5. Line 60 - 63: What do you mean "the extent of carbonate and other secondary mineral precipitation using elemental concentrations along is difficult"? Do you mean the carbonation extent of basalts? What do you mean the supply rates? Are the supply rates for the CO₂ gas and water? Why not just take the mineral sample after the injection and analyze the mineral composition and chemical compositions?
6. Line 217: what is the uncertainty of the method based on conservative tracers?
7. The uncertainty of the method based on the calcium isotopes is too large. The difference from the lowest efficiency 42% to the highest efficiency 100% is 60%!!! How can we decide that all the dissolved CO₂ gas has been completely sequestered.
8. Where did the calcium come from? Did the calcium come from the groundwater itself or from the dissolution of basalts or other minerals?
9. The paper needs to list the chemical formula of all the minerals mentioned in the context, such as zeolite, smectite?
10. Have you taken the solid samples during or after the sequestration? How did you make sure the formation of the specific minerals by the calcium isotope method, such as calcite, zeolite or smectite, which is the advantage compared to the method based on the concentration?

Reviewer #2 (Remarks to the Author):

The Authors present a novel use of the ⁴⁴Ca/⁴⁰Ca ratios to determine the extent of carbonate precipitation based on known fractionation factors - in doing so they agree, with less precision,

similar estimates based on mass balance

Reviewer #3 (Remarks to the Author):

This study NCOMMS-18-15336776 presents a new isotopic approach (Ca isotopes) for evaluating the amount of CO₂ and the rate at which CO₂ can be fixed as CaCO₃ by CarbFix type techniques, as experimented with in Icelandic basaltic aquifers. I agree that Ca isotopes are a powerful tool applicable to mineral carbonation projects for assessing the amount and rate of calcite precipitation, and therefore this study is of significant input to the growing number of studies into mineral carbonation for sequestering excess CO₂. Investigating and finding safe, effective ways of sequestering CO₂ is of immediate and important relevance.

The Ca isotopes are sourced naturally from the bedrock/aquifer system and therefore require no expensive addition of tracers.

It is also of value that this study provides a new, additional method (Ca-isotopes) for calculating the amount of CO₂ sequestered by this method, in addition to existing tracer techniques. There is reasonable agreement in the estimates of CO₂-sequestration from both methods.

This approach of using Ca-isotopes to measure the extent of calcite precipitation in a different type of natural system has already been demonstrated successfully by Owen et al. [2016]. The present study should refer to Owen et al. [2016] to strengthen their argument of using Ca-isotopes to calculate calcite-precipitation amounts and to ensure suitable referencing of existing work. In the specific case of Owen et al. [2016], Mg/Ca, Sr/Ca and Ba/Ca were also available to validate the Ca-isotope technique - i.e. there is a preexisting strong case for using $\delta^{44}\text{Ca}$ for calculating calcite precipitation amounts.

There is a figure in Owen et al. [2016] (figure 4), which provides a schematic view of Ca cycling through a cave system showing the evolution of drip-water Ca concentration, $\delta^{44}\text{Ca}$, Mg/Ca...as a result of calcite precipitation along a flow path. I mention this because when I came across it, it helped me make a connection between an entirely natural system (cave formation and speleothem formation) and these CarbFix experiments. In the natural case, the CO₂ is concentrated in the soil zone by plant activity and dissolved in soil water before dissolving bedrock and then precipitating secondary calcium carbonate. CarbFix is very similar except that the bubbling of CO₂ into water and the pumping of that water into the ~400m depth aquifer is done by engineers. Rightly/wrongly, the similarity of the two processes made me less concerned about potential negative side effects of this CarbFix approach on the aquifers. I include this simply in case it is of use to the authors.

Overall I would recommend this study for publication but believe that the reasonably large number of edits below should be implemented (or argued against) to improve the clarity, readability and robustness of the article.

Note: CaveCalc would be a useful tool for authors working on these mineral carbonation projects as it specifically models all of the major isotope systems (calcium, oxygen, carbon, including C14) during the evolution of calcite dissolution/precipitation. cf Owen et al. [2018]. In this way it could help to combine the DIC and ¹⁴C/¹²C measurements of Matter et al. [2016] with the Ca isotope measurements of the present study to allow even more robust interpretations.

The clarity and readability of the current study needs to be improved. Unlike e.g. Gislason and Oelkers [2014], Matter et al. [2016]), there is not a clear overview (early in the paper) of the mechanism by which the CO₂ is fixed as CaCO₃ in these Icelandic basaltic aquifers. This should be clarified before moving on to explain how Ca isotopes are used to quantify calcite-precipitation. In particular it is important to outline how and why CO₂-injection locally resets $\delta^{44}\text{Ca}_{\text{water}}$ to a

basalt value by dissolving basalt. It is then the increase in $d_{44}\text{Ca}$ to values higher still than the original $d_{44}\text{Ca}$ groundwater (caused by fractionation during calcite precipitation) that allow the calcite precipitation amounts to be calculated. Before reading alternative papers on the CarbFix project, my initial assumption was that the Ca-isotopes should increase above the pre-injection Ca_{44} values, which are already significantly higher than basalt values. It is perhaps not unusual for someone thinking about groundwaters (but not with CO_2 -injection) to make this incorrect assumption.

Details of the calculation/mechanism for calculating the amount CO_2 -sequestered should be included (in particular the number of moles of CO_2 converted to CaCO_3 for each mol $\text{CO}_2(\text{aq})$ injected into the aquifer). Currently only the calculation for the amount of Ca removed is detailed, I believe. If the calcite is precipitated according to the reaction $\text{Ca}^{2+} + 2\text{HCO}_3^- \rightarrow \text{CaCO}_3 + \text{H}_2\text{O} + \text{CO}_2$ then CO_2 is also produced as a result of the precipitation mechanism. Perhaps the authors can also comment on how they believe SI calcite increases subsequent to the initial post-injection decrease in SI calcite to unsaturated values. Is this purely caused by [Ca]-increase in DIC-rich solutions despite their low pH (and therefore low CO_3^{2-})? Or does it involve mixing between these [Ca]-rich solutions and surrounding aquifer solution with high pH (~9) and therefore higher carbonate ion concentrations?

It is not clear to me how mixing of altered fluids (by CO_2 -injection) and surrounding non-altered waters is affecting the Ca-isotope calcite-precipitation calculation. Or is it assumed that surrounding waters are too low in Ca to have an impact? It seems to me that such mixing could cause a decrease in the calculated amount of calcite precipitated (because measured d from equation on line 413 might be lowered as a result of the mixing).

From $^{14}\text{C}/^{12}\text{C}$ measurements in Matter et al. [2016], some of the post-injection Ca is sourced from the dissolution of preexisting calcite. How does this fit with using only the calcium isotope value of basalt for d_i (line 413)?

Supplementary figure S1 is poor compared to the geological cross-section of fig 1 Matter et al. [2016], which allows for a much better understanding of the system, namely: i) the depth of injection of the CO_2 , ii) the location and depth of boreholes HN-04 and HK-34 relative to the injection site, iii) the direction of groundwater flow. If possible, using this geological cross-section type figure would be beneficial for the present study.

Supplementary tables 1 and 2 (pre- and post-injection chemistry figures) are unfortunately fairly useless I believe. This is because it is difficult/not possible to relate the numbers in these tables to each other or to other figures in the paper or to geographical locations on the map (or geological cross section). I think this should be improved to allow interested readers to make use of these tables.

What is meant by 'shallow' or 'deep' within the context of tables 1 and 2? This is important because there are significant differences in the chemistry between the shallow and deep samples. Are all 'deep' samples from the same depth or is depth one of the factors influencing the chemistry of these measurements? What is the depth and location of these samples relative to the injection site?

I think that temperature, depth (in meters), borehole number and calculated saturation index should be included for all rows in tables 1 and 2. Temperature, depth and borehole number are currently missing for table 2 - this makes it difficult/impossible to compare pre- and post-injection measurement values. Saturation index values are missing from both tables. How were the saturation index values in e.g. Fig 3 calculated? A line or two in the supplementary methods would be useful.

Especially given the presence of Mg in solution from basalt dissolution, have the authors checked

the mineralogy of the precipitated CaCO₃ - is it really calcite or in fact aragonite? Matter et al. 2016 provides this information, but it should be made clear here too, particularly as Calcite fractionation factors are used for all calculations. If the mineralogy was not calcite then the calculations could be invalid.

Bibliography

Sigurdur R. Gislason and Eric H. Oelkers. Carbon storage in basalt. *Science*, 344(6182):373–374, 2014. ISSN 10959203. doi: 10.1126/science.1250828.

Juerg M. Matter, Martin Stute, Sandra Snæbjörnsdóttir, Eric H. Oelkers, Sigurdur R. Gislason, Edda S. Aradóttir, Bergur Sigfusson, Ingvi Gunnarsson, Holmfrídur Sigurdardóttir, Einar Gunnlaugsson, Gudni Axelsson, Helgi A. Alfredsson, Domenik Wolff-Boenisch, Kiflom Mesfin, Diana Fernandez De La Reguera Taya, Jennifer Hall, Knud Dideriksen, and Wallace S. Broecker. Rapid carbon mineralization for permanent disposal of anthropogenic carbon dioxide emissions. *Science*, 2016. ISSN 10959203. doi: 10.1126/science.aad8132.

R A Owen, C C Day, C.-Y. Hu, Y.-H. Liu, M D Pointing, C L Blättler, and G M Henderson. Calcium isotopes in caves as a proxy for aridity: Modern calibration and application to the 8.2 kyr event. *Earth and Planetary Science Letters*, 443:129–138, jun 2016. doi: 10.1016/j.epsl.2016.03.027.

Robert Owen, Christopher C. Day, and Gideon M. Henderson. CaveCalc: A new model for speleothem chemistry & isotopes. *Computers & Geosciences*, 119:115–122, oct 2018. ISSN 00983004. doi: 10.1016/j.cageo.2018.06.011.

URL <https://www.sciencedirect.com/science/article/pii/S0098300417312384> <https://linkinghub.elsevier.com/retrieve/pii/S0098300417312384>.

Reviewer #1 (Remarks to the Author):

The paper shows a new method, calcium isotope-based quantification of in-situ mineral carbonation rates. The application and development of this method is novel and will be of interest to others in the wider field (passive carbonation). However, the current-version paper is hard to understand for the readers who are interested in the in-situ and ex-situ mineral carbonation as well as passive mineral carbonation but not in the field of geology or isotope technology. I have the following suggestions and questions to clarify, detailed below:

1. The structure of this paper needs to be adjusted. The innovation in this paper is that the calcium isotope method can be used into the in-situ mineral carbonation and the quantification of CO₂ sequestration rates. The authors need to clearly clarify how this method was applied into your project and how to calculate the efficiency? We have enhanced the detail of our methods in both the text and particularly in the supplement, including all the formula used. We have also increased the description of the Ca method.

2. What are the relationships among CarbFix, CarbFix2 and CarbFix1? Is the CarbFix1 the same as CarbFix? The authors need to clarify it.

Yes, we were interchangeably using 'original CarbFix' and 'CarbFix1'. This has now been uniformly changed to 'original CarbFix'.

3. The description about the calculation based on the concentration needs to be reduced and concentrated. The focus in this paper is not to describe the method based on the change of concentration anymore. In contrast, the problems of the method based on the concentration needs to be clearly clarified and the method based on the calcium isotope method needs to be described in more details.

We've made the concentration description more concise. The isotope-based methods are described in more detail, particularly in the supplement.

4. Line 57 - 59: what are the "different minerals", such as? Why do the authors need to mention the modelling about the different minerals?

We've defined these in the revised text – basically several different carbonate and silicate secondary minerals (L57-58). Without the modelling, we would not have much of an idea into which phases the different elements are going. This is groundwater studies, so without modelling it is often impossible to ascertain exactly what is happening at several hundred metres depth.

5. Line 60 - 63: What do you mean "the extent of carbonate and other secondary mineral precipitation using elemental concentrations along is difficult"? Do you mean the carbonation extent of basalts? What do you mean the supply rates? Are the supply rates for the CO₂ gas and water? Why not just take the mineral sample after the injection and analyze the mineral composition and chemical compositions?

Clarified to say that it is difficult to determine the extent and the flux of mineral precipitation using only dissolved elemental concentrations. Using solids to calculate fluxes is not possible: firstly because changes are tiny (which is why the more greatly changing dissolved concentrations are used), and secondly, because determining fluxes via solids is also not possible. We'd be looking for a few hundred tonnes of carbonate spread over millions of m³, all several hundred metres below the surface. Clarified in L64-66.

6. Line 217: what is the uncertainty of the method based on conservative tracers?

We have re-calculated this from the data of Matter et al., 2016, and the uncertainty is $\pm 7\%$. This is now mentioned in the text. This is smaller than the estimate based on Ca isotopes, because it is based on only 2 measurements (because C is being injected), while the Ca isotope estimate is based on 19 measurements (see below), which means that the error is larger, but that time-resolution is possible.

7. The uncertainty of the method based on the calcium isotopes is too large. The difference from the lowest efficiency 42% to the highest efficiency 100% is 60%!!! How can we decide that all the dissolved CO₂ gas has been completely sequestered.

We have clarified the text to say that this is the 2sigma error, and have propagated the uncertainty a few different ways, but always get similar errors. There are two main causes of the high error: the fact that the total precipitation amount is the amalgamation of 19 separate measurements through time (and their individual errors, which have to be propagated), and (more significantly) that the experimental calibrations on the fractionation factors were done 15 years ago, and therefore their errors (which we have to include in our propagation) are high. Some groups are re-doing these calibrations, but they are not published yet. We note that the paper that Reviewer 3 made us aware of (Owen et al.), which performed a similar calculation based on cave carbonate Ca isotopes have similar errors on their f numbers.

Finally, clearly the error is large, but the advantage of the Ca isotopes is that it gives time resolution, as well as an independent method for understanding the CarbFix reactions, which after all are occurring hundreds of metres underground, and therefore otherwise difficult to quantify.

8. Where did the calcium come from? Did the calcium come from the groundwater itself or from the dissolution of basalts or other minerals?

From dissolution of the basalt. This is a basic premise of the CarbFix project – it was originally mentioned in L154-155, but has now been expanded in L99-104 and L185-187.

9. The paper needs to list the chemical formula of all the minerals mentioned in the context, such as zeolite, smectite?

We feel it would be a bit of a distraction to do this, given that literally entire papers have been written on this subject for CarbFix, mainly Aradóttir et al., 2012, but also Snæbjörnsdóttir et al., 2017 and 2018.

10. Have you taken the solid samples during or after the sequestration? How did you make sure the formation of the specific minerals by the calcium isotope method, such as calcite, zeolite or smectite, which is the advantage compared to the method based on the concentration?

Solid samples were taken both from the monitoring borehole and from the pumps – which showed that calcite was precipitating (Matter et al., 2016). This is now made clearer: L 103-104; 177. However, secondary minerals such as zeolite or smectite can only be examined through modelling (e.g. Snæbjörnsdóttir et al., 2018) – as mentioned above, finding secondary minerals by boring in such a large volume of rock is effectively impossible. This is why examination has to proceed via solution chemistry and isotopes.

Reviewer #2 (Remarks to the Author):

The Authors present a novel use of the $^{44}\text{Ca}/^{40}\text{Ca}$ ratios to determine the extent of carbonate precipitation based on known fractionation factors - in doing so they agree, with less precision, similar estimates based on mass balance.

Line 35: efficient approach

Done.

Line 39: carbonation, which...

Done.

Line 49: CarbFix

Done.

Line 51: CO₂-charged water (add reference)

Done.

Line 88: 2). In mid...

Done.

Line 136: calcite saturation (Authors to provide how this was calculated)

Done in Supplement.

Line 150: most of the dissolved Ca is taken up (In table 2 Ca concentration of pre and post injection are similar so don't easily see a decrease – please explain

This is because the flux massively increases in post-injection samples, due to active pumping. So a similar concentration, but higher water flow, results in a much higher flux. Now explained in L181-183, plus in the Supplement.

Line 158: pH recovered (Authors to provide mechanism for this recovery) Line 161 (and elsewhere in text) change ^{14}C to $^{14}\text{CO}_2$

Done for pH recovery – L173-175. Not for $^{14}\text{CO}_2$ though, as (in that particular example), it would have to be $\text{Ca}^{14}\text{CO}_3$. This generally complicates matters, so we feel it's simpler to keep as ^{14}C .

Lines 192-193 analytical uncertainty and the uncertainty in the temperature dependent fractionation factor (Authors quantify the 'uncertainty' and say which one is the largest uncertainty)

Done in supplement, including reference of a paper that uses Ca isotopes for a similar purpose, and comes to similar errors on calculation of f.

Lines 207-208: slowing of the reaction as the injected carbon dioxide (i.e. the reactants) becomes exhausted (Implies that the CO₂ is reacting directly and not CO₃²⁻ - also why would reaction rate slow if calcite saturation remains high?

The system is supersaturated for calcite prior to injection as well (Alfredsson et al., 2012). So it is likely that the system is returning to its previous state, i.e. supersaturated for calcite, but little precipitation due to the flux reasons mentioned above. We also note that the saturation calculations are a model of what can happen, rather than necessarily of what is happening, which is why direct measurements are useful.

Line 223: Ca isotopes appear to be a useful tool... (not sure I fully agree – it is complementary to the published mass balance approach (95% calcite precipitation) but gives a larger uncertainty 40 -100% calcite precipitation so I would say it's semi-quantitative at best)

Well, it's still quantitative, because it's numerical, even with a high uncertainty. But we agree that the uncertainty is relatively high (also now discussed in the Supplement). But we have changed the emphasis of the conclusions to say that the Ca isotope approach confirms the phase that the C is being taken into, and (this is key) that the isotopes provide temporal resolution of precipitation rates.

Line 246: detailed in a series of studies 21-24. Briefly,... (Authors to add a paragraph on method) Line 432 – Authors to include a worked through Precipitation calculation/precipitation rate

We already had a brief paragraph on the method before this sentence. We have added some more detail. Calculation formulae (and references for the formulae) have now been added to the Supplement (L492-520).

Reviewer #3 (Remarks to the Author):

This study NCOMMS-18-15336776 presents a new isotopic approach (Ca isotopes) for evaluating the amount of CO₂ and the rate at which CO₂ can be fixed as CaCO₃ by CarbFix type techniques, as experimented with in Icelandic basaltic aquifers. I agree that Ca isotopes are a powerful tool applicable to mineral carbonation projects for assessing the amount and rate of calcite precipitation, and therefore this study is of significant input to the growing number of studies into mineral carbonation for sequestering excess CO₂. Investigating and finding safe, effective ways of sequestering CO₂ is of immediate and important relevance.

The Ca isotopes are sourced naturally from the bedrock/aquifer system and therefore require no expensive addition of tracers.

It is also of value that this study provides a new, additional method (Ca-isotopes) for calculating the amount of CO₂ sequestered by this method, in addition to existing tracer techniques. There is reasonable agreement in the estimates of CO₂-sequestration from both methods.

This approach of using Ca-isotopes to measure the extent of calcite precipitation in a different type of natural system has already been demonstrated successfully by Owen et al. [2016]. The present study should refer to Owen et al. [2016] to strengthen their argument of using Ca-isotopes to calculate calcite-precipitation amounts and to ensure suitable referencing of existing work. In the specific case of Owen et al. [2016], Mg/Ca, Sr/Ca and Ba/Ca were also available to validate the Ca-isotope technique - i.e. there is a preexisting strong case for using d⁴⁴Ca for calculating calcite precipitation amounts.

We thank the reviewer for directing us to this paper. It is indeed useful for demonstrating the calculations, approach and uncertainties. It is now referenced throughout the text and supplement.

There is a figure in Owen et al. [2016] (figure 4), which provides a schematic view of Ca cycling through a cave system showing the evolution of drip-water Ca concentration, d⁴⁴Ca, Mg/Ca...as a result of calcite precipitation along a flow path. I mention this because when I came across it, it helped me make a connection between an entirely natural system (cave formation and speleothem formation) and these CarbFix experiments. In the natural case, the CO₂ is concentrated in the soil zone by plant activity and dissolved in soil water before dissolving bedrock and then precipitating secondary calcium carbonate. CarbFix is very similar except that the bubbling of CO₂ into water and the pumping of that water into the ~400m depth aquifer is done by engineers.

Rightly/wrongly, the similarity of the two processes made me less concerned about potential negative side effects of this CarbFix approach on the aquifers. I include this simply in case it is of use to the authors.

Again, thanks for directing us to this paper.

Overall I would recommend this study for publication but believe that the reasonably large number of edits below should be implemented (or argued against) to improve the clarity, readability and robustness of the article.

Note: CaveCalc would be a useful tool for authors working on these mineral carbonation projects as it specifically models all of the major isotope systems (calcium, oxygen, carbon, including C14) during the evolution of calcite dissolution/precipitation. cf Owen et al. [2018]. In this way it could help to combine the DIC and 14C/12C measurements of Matter et al. [2016] with the Ca isotope measurements of the present study to allow even more robust interpretations.

Another good idea that we will try to implement on future Carbfix2 studies. But, reading through CaveCalc, it looks like it might need to be adapted for the silicate weathering reactions that occur in CarbFix (see below).

The clarity and readability of the current study needs to be improved. Unlike e.g. Gislason and Oelkers [2014], Matter et al. [2016]), there is not a clear overview (early in the paper) of the mechanism by which the CO₂ is fixed as CaCO₃ in these Icelandic basaltic aquifers. This should be clarified before moving on to explain how Ca isotopes are used to quantify calcite-precipitation. In particular it is important to outline how and why CO₂-injection locally resets d⁴⁴Ca water to a basalt value by dissolving basalt. It is then the increase in d⁴⁴Ca to values higher still than the original d⁴⁴Ca groundwater (caused by fractionation during calcite precipitation) that

allow the calcite precipitation amounts to be calculated. Before reading alternative papers on the CarbFix project, my initial assumption was that the Ca-isotopes should increase above the pre-injection Ca44 values, which are already significantly higher than basalt values. It is perhaps not unusual for someone thinking about groundwaters (but not with CO₂-injection) to make this incorrect assumption.

We agree and have added more detail on both Carbfix and the mechanism of CO₂ sequestration. This is now explained in L45-49, L117-123 and L202-206.

Details of the calculation/mechanism for calculating the amount CO₂-sequestered should be included (in particular the number of moles of CO₂ converted to CaCO₃ for each mol CO₂(aq) injected into the aquifer). Currently only the calculation for the amount of Ca removed is detailed, I believe. If the calcite is precipitated according to the reaction $\text{Ca}^{2+} + 2\text{HCO}_3^- \rightarrow \text{CaCO}_3 + \text{H}_2\text{O} + \text{CO}_2$ then CO₂ is also produced as a result of the precipitation mechanism. Perhaps the authors can also comment on how they believe SI calcite increases subsequent to the initial post-injection decrease in SI calcite to unsaturated values. Is this purely caused by [Ca]-increase in DIC-rich solutions despite their low pH (and therefore low CO₃²⁻)? Or does it involve mixing between these [Ca]-rich solutions and surrounding aquifer solution with high pH (~9) and therefore higher carbonate ion concentrations?

We've added the equation (and more detail) in L521-536. The equation is not quite as suggested: unlike Owen et al, this is a silicate weathering reaction, rather than a carbonate weathering one. It also takes place at different pH, without the ocean carbonate cycle.

The decrease in calcite SI is due to rapid basalt dissolution, as well as other factors, now detailed in L119-125.

It is not clear to me how mixing of altered fluids (by CO₂-injection) and surrounding non-altered waters is affecting the Ca-isotope calcite-precipitation calculation. Or is it assumed that surrounding waters are too low in Ca to have an impact? It seems to me that such mixing could cause a decrease in the calculated amount of calcite precipitated (because measured $d_{44}\text{Ca}$ from equation on line 413 might be lowered as a result of the mixing). While the Ca concentrations of fluids with and without added CO₂ are similar, the flow rates (and hence fluxes) change dramatically, due to active pumping. So, pre-injection values are not factored in as they are negligible in the calculations presented in the manuscript. This is now explained better in L198-200, and the supplement.

From 14C/12C measurements in Matter et al. [2016], some of the post-injection Ca is sourced from the dissolution of preexisting calcite. How does this fit with using only the calcium isotope value of basalt for $d_{44}\text{Ca}$ (line 413)?

We now explain this in the Supplement (L481-483). Basically, dissolution of calcite will decrease the $d_{44}\text{Ca}$ of the solution – which means that the calculations result in f changing, and determining that no carbonate is being precipitated. This is shown in the figures: calcite might only be dissolving immediately after injection when pH decreases. During these times, the Ca isotope-derived rates are effectively 0.

Supplementary figure S1 is poor compared to the geological cross-section of fig 1 Matter et al. [2016], which allows for a much better understanding of the system, namely: i) the depth of injection of the CO₂, ii) the location and depth of boreholes HN-04 and HK-34 relative to the injection site, iii) the direction of groundwater flow. If possible, using this geological cross-section type figure would be beneficial for the present study.

More figures from previous Carbfix studies have been added to the Supplement.

Supplementary tables 1 and 2 (pre- and post-injection chemistry figures) are unfortunately fairly useless I believe. This is because it is difficult/not possible to relate the numbers in these tables to each other or to other figures in the paper or to geographical locations on the map (or geological cross section). I think this should be improved to allow interested readers to make use of these tables.

In response to this comment we have added more cross sections and maps to make determining the locations easier.

What is meant by 'shallow' or 'deep' within the context of tables 1 and 2? This is important because there are significant differences in the chemistry between the shallow and deep samples. Are all 'deep' samples from the same depth or is depth one of the factors influencing the chemistry of these measurements? What is the depth and location of these samples relative to the injection site?

Well depths are not possible to give, beyond shallow and deep: the deep wells are encased to below 400m, whereas the shallow wells draw water from above that. But the system is such that specific depths are not known. We have added this detail to L562-565.

I think that temperature, depth (in meters), borehole number and calculated saturation index should be included for all rows in tables 1 and 2. Temperature, depth and borehole number are currently missing for table 2 - this makes it difficult/impossible to compare pre- and post-injection measurement values. Saturation index values are missing from both tables.

This is because Table 2 (the post-injection data) only reports data from a single monitoring hole, and so location and temperature are constant. We have now made this clear in the caption.

How were the saturation index values in e.g. Fig 3 calculated? A line or two in the supplementary methods would be useful.

Added in the methods section (L301-304).

Especially given the presence of Mg in solution from basalt dissolution, have the authors checked the mineralogy of the precipitated CaCO₃ - is it really calcite or in fact aragonite? Matter et al. 2016 provides this information, but it should be made clear here too, particularly as Calcite fractionation factors are used for all calculations. If the mineralogy was not calcite then the calculations could be invalid.

Good point – we've added the statement that in material recovered from pumps and the single sample borehole only calcite was determined (L117-123), originally mentioned in Snæbjörnsdóttir et al., 2017.

Reviewers' comments:

Reviewer #1 (Remarks to the Author):

Thanks to the authors' careful revision, the current manuscript is easier to understand and presents the obvious innovations. It has been stated that the manuscript shows a novel method based on calcium isotope for quantification of in-situ mineral carbonation rates. However, the high uncertainty is the key concern. It would be acceptable if the authors can decrease the uncertainty to a suitable extent.

1. Since the data of Matter et al., 2016 shows the much smaller uncertainty, $\pm 7\%$, why do you think the method based on the calcium isotope to quantify the carbonation rates is better than their method, which has the uncertainty $\pm 30\%$? You outline that your method has the specific advantage of time resolution. Have you clearly demonstrated that their method cannot be used for the time-resolution calculation? It is very important because the readers can easily compare your work and theirs and ask why they need to choose yours. It is not clear that the calculation method in the work of Matter et al., 2016 cannot be used with time.

2. Why do you mean "using concentration of solids from drill cores would also be extremely difficult, due to the relatively small amount of solids precipitated over millions of m³ of aquifer"? Have you read the recent paper, "Wang, F. et al. Quantifying kinetics of mineralization of carbon dioxide by olivine under moderate conditions. Chemical Engineering Journal 360, 452-463 (2019)"? In their calculation method, the calculation of mineral carbonation efficiency is not related to the amount or volume of aquifer. In contrast, once the carbon content during carbonation is analyzed, the time-resolution carbonation efficiency can be calculated, which is the analysis result you can obtain once you have taken the solid sample from the boreholes. It is known the newest LECO CS844 instrument can analyze the total carbon with precision 0.5% RSD or 0.0003 mg. It seems like the method may be still suitable for the process with relatively small amount of solid precipitated and the system error according to their method may be smaller than that in your work. Please explain.

3. You have figured out that there are two main causes of the high error in your manuscript. Especially, the majority is due to the experimental calibrations on the fractionation factors. Can you address it firstly, or try your best to decrease the uncertainty in your method to a reasonable extent? It is not fair enough that you need to have a similar error to the others' work (Owen et al.). Otherwise, as Reviewer #2 suggested, the semi-quantitative is much better to be called for this method. It is much less attractive with such high uncertainty.

Reviewer #2 (Remarks to the Author):

Authors need to state why a Rayleigh model was applicable over other models (e.g. total reflux - Galimov 2006 Organic Geochemistry 37, 1200-1262 Figure 4.

Reviewers' comments:

Reviewer #1 (Remarks to the Author):

Thanks to the authors' careful revision, the current manuscript is easier to understand and presents the obvious innovations. It has been stated that the manuscript shows a novel method based on calcium isotope for quantification of in-situ mineral carbonation rates. However, the high uncertainty is the key concern. It would be acceptable if the authors can decrease the uncertainty to a suitable extent.

1. Since the data of Matter et al., 2016 shows the much smaller uncertainty, $\pm 7\%$, why do you think the method based on the calcium isotope to quantify the carbonation rates is better than their method, which has the uncertainty $\pm 30\%$? You outline that your method has the specific advantage of time resolution. Have you clearly demonstrated that their method cannot be used for the time-resolution calculation? It is very important because the readers can easily compare your work and theirs and ask why they need to choose yours. It is not clear that the calculation method in the work of Matter et al., 2016 cannot be used with time.

These comments prompted us to delve into the error propagation some more. We consulted with a mathematical expert in error propagation (now also acknowledged). One clear issue (and hence it is lucky the reviewer brought this up again!) was that there was a parenthesis error in our calculations ($\wedge 2$ was being applied to n as well as σ).

Through the discussions we have also changed two other aspects of the error propagation: 1) we use 2σ rather than $2s$ errors for the Ca isotope propagation. This is more correct, since we are propagating the uncertainty on each individual sample measurement, rather than the external uncertainty (i.e. long-term reproducibility, which we were applying to every sample beforehand). 2) we have also changed the error propagation formulae (for our data, but also the Matter data) to better take into account the difference in time-steps of sampling. These formulae are now given in the Supplement.

For the Ca isotope estimate, this now changes the precipitated calcite to $165 \pm 8.3 \text{ t}$ – so the uncertainty is drastically reduced by this method to $\pm 5\%$.

The same calculations also reduce the error on the Matter data to $\pm 4\%$ based on DIC, and $\pm 3\%$ based on ^{14}C . The uncertainty of these data has shrunk less, because we now also take into account the uncertainty on the model used by Matter et al., while for the Ca isotopes, the uncertainty in the model (the fractionation factor) was already inherent in the final numbers.

In terms of the difference between the approach of Matter and our approach, fundamentally, using ^{14}C or DIC tells you about the injected carbon, while using Ca isotopes tells you about basalt-sourced calcium. Hence, while it is possible to calculate a time-series from the Matter data (although not actually done in that paper – effectively you get a % of removed C with time, and can convert that into a rate), you don't get a precipitation rate, you get a carbon removal rate – and also only if you factor in a mixing model based on a different conservative tracer.

In a sense, both a carbon removal and a calcite precipitation rate are useful – albeit that the carbon removal rate does not give any direct information into which phases the missing C is going. We have clarified this in the text (L213-227). The further advantage of using Ca is that

tracers don't need to be injected to gain rates. In many scenarios, this may not be possible or is too expensive (as mentioned in the Conclusions and L208).

2. Why do you mean "using concentration of solids from drill cores would also be extremely difficult, due to the relatively small amount of solids precipitated over millions of m³ of aquifer"? Have you read the recent paper, "Wang, F. et al. Quantifying kinetics of mineralization of carbon dioxide by olivine under moderate conditions. Chemical Engineering Journal 360, 452-463 (2019)"? In their calculation method, the calculation of mineral carbonation efficiency is not related to the amount or volume of aquifer. In contrast, once the carbon content during carbonation is analyzed, the time-resolution carbonation efficiency can be calculated, which is the analysis result you can obtain once you have taken the solid sample from the boreholes. It is known the newest LECO CS844 instrument can analyze the total carbon with precision 0.5% RSD or 0.0003 mg. It seems like the method may be still suitable for the process with relatively small amount of solid precipitated and the system error according to their method may be smaller than that in your work. Please explain.

Assuming a fairly narrow groundwater flow path along the 500m separating the injection and the sampling points, there is about 3.8e8 t basalt between them. We are precipitating around 200t of carbonate (~24 t of carbon). If that carbon is distributed uniformly along that flowpath, then the C content of the rocks would only increase by 6.46e-6 %, which at best is 0.00006mg C per g of rock. It is more likely that precipitation is not uniform, but is concentrated somewhere along the flowpath – in which case, even drilling several boreholes are extremely unlikely to find the location of that carbonate, even were we able to detect any change in the C contents of the rocks.

The Wang paper, while quite neat, does require being able to measure a difference in the C content and the rock weight before and after mineralisation (e.g. Equ 13), neither of which is possible in a natural groundwater system like ours. We highlight this, and reference this paper, in Lines 66-67.

3. You have figured out that there are two main causes of the high error in your manuscript. Especially, the majority is due to the experimental calibrations on the fractionation factors. Can you address it firstly, or try your best to decrease the uncertainty in your method to a reasonable extent? It is not fair enough that you need to have a similar error to the others' work (Owen et al.). Otherwise, as Reviewer #2 suggested, the semi-quantitative is much better to be called for this method. It is much less attractive with such high uncertainty. We cannot change the calibration because this is fixed by the experiments in the publications. But as described above, using a different approach to error propagation does yield a different result.

Reviewer #2 (Remarks to the Author):

Authors need to state why a Rayleigh model was applicable over other models (e.g. total reflux - Galimov 2006 Organic Geochemistry 37, 1200-1262 Figure 4.

As explained in the Galimov paper, when total reflux occurs, there is no isotope fractionation: “the isotopic composition of the entering and outgoing materials are equal”. Total reflux is largely concerned with the composition of an intermediate product, which we don’t have in our study, as effectively Ca in basalt is turned into Ca in calcite. There is the alternative to Rayleigh fractionation, which is equilibrium fractionation (cf. Isotope Geology textbooks, such as Faure and Mensing, or White). However, as we say in Lines 163-164, this is not possible, as this mode of fractionation has a maximum amount of fractionation, which is less than what we observe here.

REVIEWERS' COMMENTS:

Reviewer #1 (Remarks to the Author):

Thanks for the authors' careful revisions. Now the key concern has been addressed and the manuscript has clearly shown the advantages and innovations. I have no further concern. The current version is more attractive and easier to be understood and is suitable to be accepted.